# Triboelectric Nanogenerators for Ocean Wave Energy Harvesting: Unit Integration and Network Construction

**Xi Liang [1,2], Shijie Liu [1,2], Hongbo Yang [1,3] and Tao Jiang [1,2,*]**

1   CAS Center for Excellence in Nanoscience, Beijing Key Laboratory of Micro-Nano Energy and Sensor, Beijing Institute of Nanoenergy and Nanosystems, Chinese Academy of Sciences, Beijing 101400, China
2   School of Nanoscience and Technology, University of Chinese Academy of Sciences, Beijing 100049, China
3   College of Engineering, Zhejiang Normal University, Jinhua 321004, China
*   Correspondence: jiangtao@binn.cas.cn

**Abstract:** As a clean and renewable energy source with huge reserves, the development of ocean wave energy has important strategic significance. Harvesting ocean wave energy through novel triboelectric nanogenerators (TENGs) has shown promising application prospects. For this technology, the integration of TENG units is the crucial step to realize large-scale network commercialization. All aspects of the TENG networking process are systematically summarized in this review, including the topology design and the circuit-connection scheme. Advancing the research on the large-scale TENG network is expected to make great contributions to achieve carbon neutrality.

**Keywords:** blue energy; triboelectric nanogenerator; network; carbon neutrality

## 1. Introduction

Facing the pressures of energy shortages and environmental deterioration, traditional fossil energy is no longer suitable considering the new directions taken by human development [1,2]. Research into clean and renewable energy is an effective approach to solve energy problems and ensure energy safety. Ocean wave energy, which has a series of advantages, is regarded as an important new energy source to be exploited [1,3–5]. Above all, the energy reserves of ocean wave energy is huge. Around the coastline worldwide, it has been estimated to be 2-3 TW [6]. Furthermore, different from other energy sources, ocean wave energy can be harvested without light, wind, or other special working conditions, so all-weather work can be achieved [4,7–9]. In addition, the exploitation of ocean wave energy does not require occupying land; therefore, many potential issues are avoided. Currently, the commercialization of ocean wave energy has been promoted to a strategic position by many counties, and will deeply influence the balance of global energy, economics, and politics.

During the past few decades, some small water-wave-energy-harvesting devices have been put into operation, which are mainly used for navigation label lights, buoys, etc. [10] Unfortunately, no present devices can reach 1-10 MW output power, and massive commercial wave energy farms are still in the conception stage [10]. The key factor restricting the scale of water wave energy harvesting is that the core components of the current devices are basically electromagnetic generators (EMGs) [11,12]. The working mode of EMGs determines the structures must include heavy magnets, metal coils, and other related components, which are bulky, expensive, and easy to corrode [13–15]. Moreover, the EMGs cannot reach the best working states under low-frequency ocean waves (<5 Hz) [11,12,16]; therefore, the energy-conversion efficiency is difficult to further break through [3]. The triboelectric nanogenerator (TENG; also called a Wang generator), invented by Wang in 2012 [17], has a completely different working principle from EMGs. TENG is based on the coupling of triboelectrification and electrostatic induction, which is the application of Maxwell's displacement current in energy and sensors [18]. Due to the unique working

mechanism, TENGs have some advantages over EMGs, such as being light weight [19–21], low cost [22–25], having a wide material-selection range [26–29], and low-frequency adaptation [30–33]. After 10 years of development, TENGs have exhibited great capabilities in harvesting mechanical energy from various sources, providing a new effective strategy for ocean wave energy exploitation [29].

In the research on harvesting ocean wave energy by TENGs, apart from the water-solid-based TENGs [12,34], most works focus on fully enclosed TENG devices. Researchers have gradually improved the output performances of TENGs, and their maximum peak power density now reaches 30.24 W/m$^3$ [35]. However, the sizes of the TENGs are generally concentrated in a few to tens of centimeters, which are not adaptable to the vast ocean surface. In 2014, the concept of harvesting water wave energy using TENG networks was first proposed [36], with TENG networking becoming the first necessary step for ocean wave energy to enter large-scale commercialization. Therefore, works solving problems in TENG unit integration and networking have great research value.

In this paper, the recent research advances in TENG unit integration and networking for water wave energy harvesting are summarized. This review covers a series of topics in the evolution from the single TENG unit to the large-scale network containing the topological organization of the TENG devices, and the further networking of point-absorption units and circuit-connection schemes. These research works have discovered and solved various possible problems related to constructing a large-scale TENG network; therefore, they have great significance for practical applications. Lastly, the existing problems and further research directions of TENG network development are also discussed in this paper.

## 2. Integration of TENG Units in Point-Absorption Device

### 2.1. Topology Structure of TENG Units Integrated in Point-Absorption Device

Point-absorption devices are the most basic unit of the ocean wave energy harvesting network; however, they usually contain more than one TENG unit inside their internal space. Their topological organization of TENG units, adaptability in water waves, spatial utilization, flexible expansion of structure, and improvements in power density are their most considered factors. From different perspectives, researchers have designed various organization methods to integrate TENG units in a point-absorption device.

#### 2.1.1. Stacked Structure

A direct organization method is used to design stacked structures, which can greatly improve the spatial utilization of the point-absorption devices. Figure 1a shows a folded structure with five contact-separation TENG units integrated inside a spherical shell—an expression of the stack structure—which was designed by Xiao et al. in 2018 [37]. When triggered by water waves, there is a phase difference between the outside shell and the inner TENG, leading to the contact-separation movement of each unit and the electrical output. More directly, Xi et al. stacked six TENG units in a cylindrical buoy through springs, achieving the high utilization rate of the internal space (Figure 1b) [38]. Driven by the wave energy and assisted with the springs, the mass inside the buoy can drive the TENG units to work and output electrical energy in cycles. In the same year, the tower-like device containing several TENG units made of polytetrafluoroethylene (PTFE) balls and arc electrodes coated with nylon film was reported (Figure 1c) [39]. Under the excitation of the external wave, PTFE balls can periodically roll back and forth on the arc surface to produce an alternating current by utilizing the TENG's freestanding mode. Stacked structures have been widely applied in the design of the point-absorption devices, and Figure 1d,e exemplifies two other stacked structures [40,41].

#### 2.1.2. Multidirectional Structure

Since the TENG units in the stack structure are only organized in one certain direction, these devices cannot collect wave energy in other directions efficiently. In 2018, Jung et al. fabricated a series of tubular TENG units (Figure 1f) [42]. For each TENG unit, when the

PTFE bar moves backward inside the polystyrene (PS) tube, the current flows between two electrodes. Then, these units were organized rotationally upward, improving the adaptation with the multidirectional water waves. Further, Jiang et al. reported a spherical device containing six multilayered TENG units symmetrically located in different directions (Figure 1i) [43], and the orientation of each TENG unit is shown in Figure 1j. Regardless of the water wave direction, these TENG units can ensure that the entire spherical device's work and output power. The profiles of the current, the voltage, and the power of the device when it presents different orientation angles with the wave triggering direction were summarized in this work, which are centrosymmetric with respect to the angle of 45°. The result illustrates that the multidirectional organization method ensures efficient energy conversion under irregular water waves. Similarly, inspired by lotus flowers, Wen et al. organized six multilayered folded TENG units in the form of petals, and set two multi-arched TENG units at the flower stamen (Figure 1g) [44]. Triggered by water waves, the device converts kinetic energy into electric energy through the "flowering" and "folding" motions of the TENG units. The design of organizing TENG units in various directions has been widely developed, and Figure 1h,k exhibit other two similar structures [9,45].

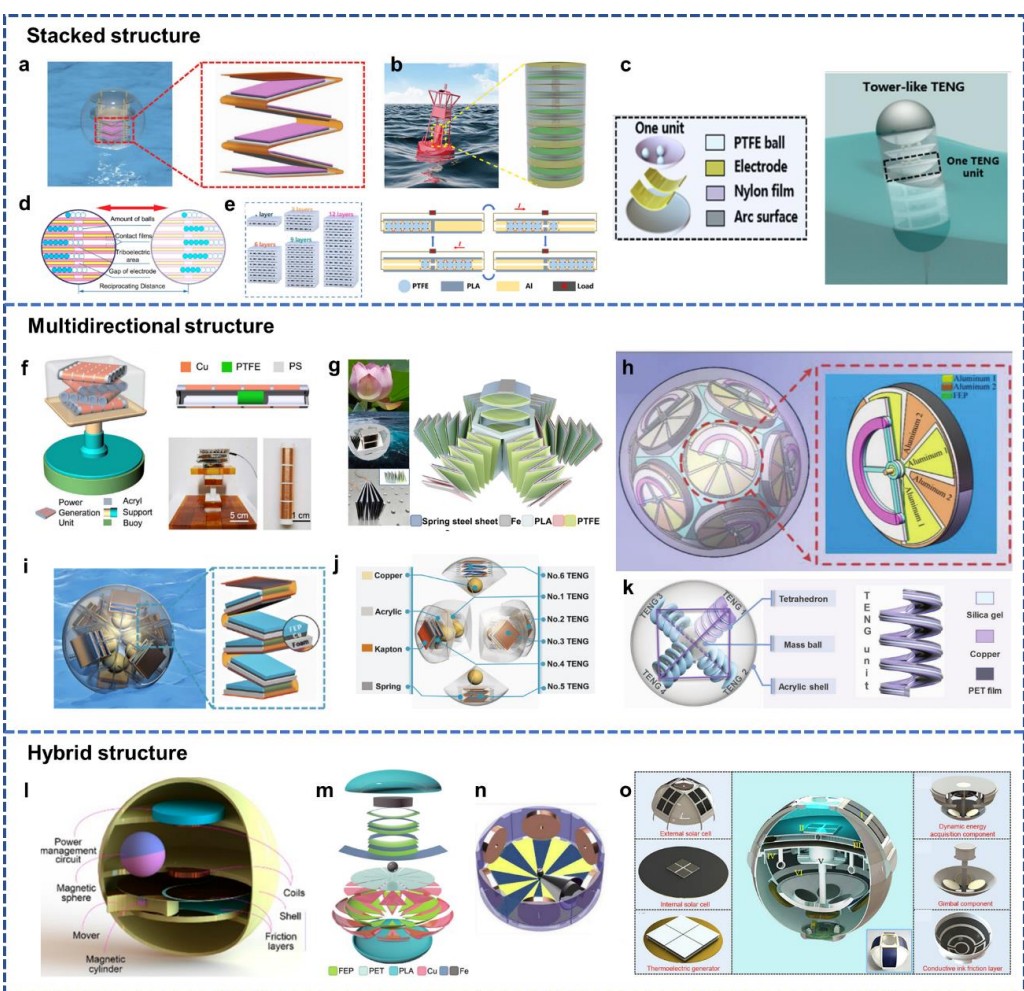

**Figure 1.** (**a**) Schematic diagram of the folded structure with five TENG units inside the spherical device floating on water. Reprinted with permission from ref. [37], Copyright 2018, Wiley. (**b**) Structure of the stacked TENG in the buoy. Reprinted with permission from ref. [38], Copyright 2019, Elsevier. (**c**) Schematic diagram of the tower-like device containing several TENG units. Reprinted with permission from ref. [39], Copyright 2019, American Chemical Society. (**d**) Spherical device with multiple TENG units stacked inside it. Reprinted with permission from ref. [40], Copyright 2021, American Chemical Society. (**e**) Schematic working cycle of a single TENG unit, and installing the device

by stacking the multiple layers. Reprinted with permission from ref. [41], Copyright 2022, the authors. (**f**) Device with a series of tubular TENG units organized rotationally upward. Reprinted with permission from ref. [42], Copyright 2018, Elsevier. (**g**) Schematic diagram of the device structure inspired by an unopened bud and a blooming lotus. Reprinted with permission from ref. [44], Copyright 2021, Elsevier. (**h**) Structure model of the spherical device with multiple TENG units integrated in various directions, and the enlarged view of one TENG unit. Reprinted with permission from ref. [45], Copyright 2022, Wiley. (**i**) Spherical structure with six symmetrically distributed multilayered TENG units, as well as the structure of the internal multilayered TENG. (**j**) The orientation of each TENG unit. (**i,j**) Reprinted with permission from ref. [43], Copyright 2019, Royal Society of Chemistry. (**k**) Structure of the spherical TENG with four spiral TENG units inside. Reprinted with permission from ref. [9], Copyright 2022, Wiley. (**l**) Structure of the hybrid triboelectric-electromagnetic water wave energy harvester. Reprinted with permission from ref. [46], Copyright 2019, American Chemical Society. (**m**) Breakdown drawing of the oblate spheroidal device with TENG units and EMG units. Reprinted with permission from ref. [47], Copyright 2019, Wiley. (**n**) Structure of the cylindrical hybrid nanogenerator. Reprinted with permission from ref. [48], Copyright 2020, Elsevier. (**o**) Diagram of the hybrid nanogenerator with piezoelectric, electromagnetic, photovoltaic, and thermotropic units. Reprinted with permission from ref. [8], Copyright 2022, Science China Press and Springer-Verlag GmbH Germany, part of Springer Nature.

### 2.1.3. Hybrid Structure

In order to achieve higher energy-conversion efficiency, some researchers integrated multiple power-generation modes with TENG units in one point-absorption device. In 2019, Wu et al. reported a hybrid triboelectric-electromagnetic water wave energy harvester (Figure 1l), which contains an EMG unit in one layered space of the spherical structure and a TENG unit in another layered space [46]. In water waves, the magnetic sphere of the EMG unit not only provides a variable magnetic field to the coils, but also drives the freestanding TENG unit underneath it to work. Figure 1m,n show two other similar integration schemes of TENG units with EMG units, which were designed according to the structural features of the point-absorption devices [47,48]. More comprehensively, Xue et al. proposed an energy harvester integrating TENG units with piezoelectric, electromagnetic, photovoltaic, and thermotropic units to fully collect ocean energy, including water wave energy (Figure 1o) [8]. In the TENG unit, the PTFE film was attached to the surface of the sliding part as the negative triboelectric material, and the interdigital electrode served as the electrode and the positive friction layer. Through organizing multiple power-generation modes, the space that is difficult to be occupied by the TENG units can be effectively utilized, and the energy conversion is improved.

For large-scale TENG networks, the TENG unit integration in one point-absorption device is the most fundamental step. At this stage, the most extensive topological organization method is arranging TENG units in one direction or in multiple directions, and some researchers combined other power-generation modes with the TENG units. These methods have solved some problems; however, for the commercialization of the TENG networks, their adaptability to ocean waves must be further discussed.

### 2.2. Circuit Connection of TENG Units Integrated in Point-Absorption Device

To integrate multiple TENG units in a point-absorption device, not only the spatial arrangement needs to be considered, but also the circuit connection. Driven by irregular water waves, the output power of each TENG unit inside the device cannot be completely superimposed. Through designing reasonable circuit-connection schemes, researchers minimized the power loss in the integration process of the TENG units.

### 2.2.1. Direct Connection

For some special structures, the motion state of each TENG unit in the point-absorption device remains almost the same under the water wave triggering. Taking the device in

Figure 1a as an example, the multiple TENG units of the folded structure can contact and separate synchronously because they are subjected to the force of the same mass block. In this case, the TENG units were directly connected in parallel. Figure 2a,b shows the output performances of the device when the number of TENG units increases from three to seven, and the peak values of the output current and transferred charge proportionally increase with the unit number [37]. This result illustrates that the direct parallel connection mode of the TENG units in this structure is reasonable. For the tower-like device in Figure 1c, the direct parallel connection mode was also applied due to all PTFE balls moving in the same phase, as shown in Figure 2c [39].

However, the simple parallel connection mode is only suitable for TENG units that work synchronously. If this condition cannot be met, the connection mode is not appropriate. In Figure 2d, the spherical device with the spring-assisted swing structure contains four TENG units, two of which work in the same phase, while the other two are completely in reverse [49]. According to the working phase, these units were divided into two sets. Inside the sets, the TENG units were directly connected in parallel; however, between the sets, the opposite electrodes were further connected (Figure 2e). The special connection mode was designed by the characteristics of the structure, which is not universal. In most instances, under irregular water waves, the movements of TENG units are disordered, so the direct connection will cause the offsetting of positive and negative electrical signals.

### 2.2.2. Connection through Rectifier Bridges

In order to avoid energy loss, rectifier bridges are applied in many devices. For example, the six TENG units in the device shown in Figure 1i were each first connected to rectifier bridges, and then connected together (Figure 2f) [43]. After connecting to rectifier bridges, the two situations of parallel and series were discussed, showing that the parallel connection can improve the output current, while the series connection can improve the output voltage. The results illustrate that the introduction of rectifier bridges effectively reduces the energy loss. Moreover, for the point-absorption devices combining other power-generation modes with TENG units, the rectification step is more important. Figure 2g shows a hybrid nanogenerator gathering a TENG unit, a PENG unit, and an EMG unit, which was reported by Wu et al. in 2022 [50]. Due to the different working modes and output characteristics, these units were connected in parallel through three rectifier bridges to power the energy-storage module. In these situations, connecting TENG units through rectifier bridges is a simple and effective solution.

### 2.2.3. Connection through Charge-Excitation Scheme

Compared with connecting TENG units directly or through rectifier bridges, designing charge-pumping or energy-excitation schemes for the point-absorption device is a better solution. In 2020, Xu et al. demonstrated a high-performance TENG based on charge shuttling, and applied it to the device for water wave energy harvesting [35]. Figure 2h is the specific structure of this device, which contains four main TENG units and two pump TENG units. According to the different action phases of the units, they were divided into two groups: the left pump TENG drives the main TENG units L1 and L2 as the left phase, and the right pump TENG drives the main TENG units R1 and R2 as the right phase. Triggered by water waves, the slider moves smoothly along the rods and controls the main TENG units in the contact state when the corresponding pump TENG units act, facilitating charge injection. Utilizing the charge shuttling scheme, the maximum output power of the device reached 126.67 mW (Figure 2i), and the power density was 30.24 W/m$^3$, which was much higher than other general point-absorption devices.

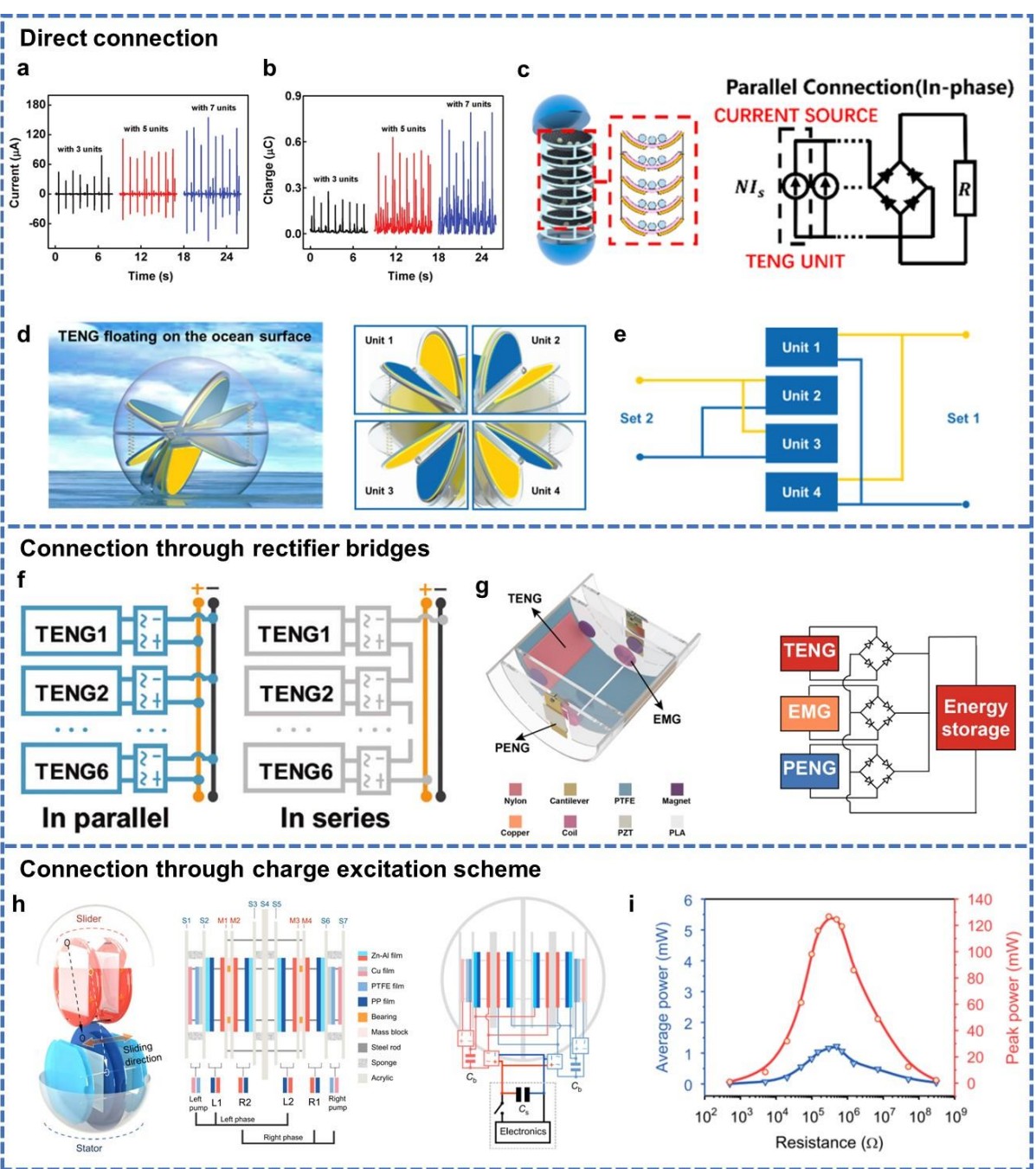

**Figure 2.** (**a**) Output current and (**b**) transferred charge of the spherical device with the number of the units in the folded structure increasing from three to seven. (**a**,**b**) Reprinted with permission from ref. [37], Copyright 2018, Wiley. (**c**) Direct parallel connection mode of the tower-like device. Reprinted with permission from ref. [39], Copyright 2019, American Chemical Society. (**d**) Working phase and (**e**) circuit-connection mode of the four TENG units inside the spherical device with spring-assisted swing structure. (**d**,**e**) Reprinted with permission from ref. [49], Copyright 2021, Elsevier. (**f**) Schematic diagrams for the parallel and series connections between the multilayered TENG units. Reprinted with permission from ref. [43], Copyright 2019, Royal Society of Chemistry. (**g**) Structure and circuit connection of the hybrid nanogenerator gathering a TENG unit, a PENG unit, and an EMG unit. Reprinted with permission from ref. [50], Copyright 2022, Elsevier. (**h**) Specific structure of the high-performance device based on charge shuttling. (**i**) Peak power and average power of the integrated device with various loads in water waves. (**h**,**i**) Reprinted with permission from ref. [35], Copyright 2020, the authors.

For the circuit connection of the TENG units in the point-absorption devices, researchers have not conducted a systematic study. In most works, the TENG units are connected together directly or through the rectifier bridges. Some researchers constructed a charge compensation system to improve the output performances while integrating TENG units. Research in this direction is of great significanc for developing high-performance and large-scale TENG networks for blue energy.

## 3. Integration of TENG Units in TENG Network

### 3.1. Topology Structure of TENG Network for Water Wave Energy Harvesting

Facing the vast ocean, the energy collection of a single point-absorption device is limited. Therefore, linking point-absorption devices into networks is necessary to increase the scale of ocean wave energy harvesting. In the process of networking, the topology structure design has great impact on the output performances. At present, researchers have developed a series of different methods to build TENG networks, improving the wave-energy-conversion efficiency.

#### 3.1.1. One-Dimensional Chain Structure

Among various TENG network structures, the simplest is to link point-absorption units along one direction to form a chain. In 2018, Zhang et al. presented a novel design based on the Pelamis snake energy harvester, which is the first offshore energy harvester to generate electricity into the grid [51]. The sea snake based on TENG units is shown in Figure 3a,b, and the TENG units were connected into a chain through the direction of ocean waves. As a wave passes through, the structure is able to flex and bend easily due to the lightweight elements of the TENG units. Similarly, the anaconda-shaped spiral multilayered TENG developed by Wang et al. also adopted the same integration method [52]. The anaconda-shaped structure allows the TENG units to adapt to different frequencies and shapes of waves, enhancing their wave-interception capability. In addition, Bai and co-workers demonstrated a high-performance tandem disk TENG for water wave energy harvesting, and the device was designed to be connected into strings (Figure 3c) [53]. The authors systematically investigated the influence of the connection point's position on the overall output performance. The results indicated that the connection at the bottom is much better than in the middle connection because the symmetric configuration of the shell makes it difficult to be pushed by waves. In addition to the direction of water waves, the chain structure can also be extended vertically to develop the underwater space. Conducting work on the tower-like TENG device (Figure 1c), Xu et al. conceived an innovative network that connected TENG units vertically to the bottom of the ocean, which is depicted in Figure 3d [39]. The acrylic shells of the devices effectively minimized dielectric shielding, ensuring the feasibility of the underwater TENG network.

#### 3.1.2. Two-Dimensional Network Structure

Compared with one-dimensional chain connections, linking the TENG units into two-dimensional networks on the seawater plane is a more widely studied solution. In 2015, Chen et al. reported a quadrilateral network design made of TENG units [54]. Figure 3e shows the schematic illustration of the configuration, which is a multilayer connection method. In the foot layer, thousands of single units were first connected together to form a community, and then in the upper layer, thousands of communities were further connected to obtain enhanced output performance. Since this scheme is simple and effective, many of later works follow this solution. Figure 3f,g lists two TENG networks constructed by the square organization method [42,55]. Different from square networks, Jiang et al. demonstrated a hexagonal TENG network in 2019 (Figure 3h) [56]. The network contains seven TENG units: one unit located in the center of the hexagon, and the other six units locates at the apexes of the hexagon. In this work, the influences of the water wave conditions on the output performance of TENG network were systematically

investigated. The highest outputs of the network reached 270 µA, 354 V, and 12.20 mW, with a corresponding power density of 3.33 W/m$^3$.

The one-dimensional chain structure and the two-dimensional network structure have their own advantages and disadvantages. For the one-dimensional chain structure, the TENG units can be arranged according to the propulsion direction of the ocean waves, making them easier to adapt to the waves. They can be also extended to deep positions into the seabed, reducing their footprint on the sea surface. However, the one-dimensional chain structure is difficult to expand to a large scale. On the contrary, the two-dimensional network structure can easily integrate numerous TENG units on the sea surface to achieve large-scale ocean wave energy harvesting. The disadvantage is that the adaptability of the two-dimensional structure to real ocean waves is unclear; therefore, the influence and interaction between each TENG unit needs to be further studied.

### 3.1.3. Mechanical Connection

In addition to the arrangement method, the mechanical connection between the TENG units in the network also plays a crucial role. Different mechanical connection modes result in different motion patterns of the TENG units, affecting the corresponding output performances. In 2018, Xu et al. proposed three types of connections (Figure 3i): a rigid plate connection, which will not deform; an elastic strip connection, which can deform to some extent; and a string connection, which can freely deform without extension [57]. Their experiments indicated that the string connection and the elastic strip connection, which can be regarded as flexible connections, are better networking strategies than the rigid one, because the rigid connection imposes too much internal constraint between units. In 2019, they further reported a self-assembly TENG network (Figure 3j) [58]. By designing rational self-adaptive magnetic joints, the network has the capabilities of self-assembly, self-healing, and facile reconfiguration, greatly improving the autonomy and robustness of the system. Moreover, Liao et al. applied the pivot hinge as the center of the link to design a hybrid system, which is exhibited in Figure 3k [59]. Each unit is supported on a bracket and then connected by two cantilever beams, which are anchored on the bearing support in the pivot hinge. The pivot hinge floats on water with polyfoam, so that the units can operate parallel to the wave direction and effectively ride the waves. In 2020, Liu et al. demonstrated a unique TENG network based on plane-like power cables consisting of spring steel tapes and three polymer films, which is shown in Figure 3l [60]. This work is the first one in which the cables are designed as connection elements and power-generation components at the same time.

In summary, the most studied current TENG networks are one-dimension chain structures and two-dimension plane structures. The chain structures generally extend towards the direction of water waves or underwater, and most of the plane structures are quadrilateral networks and hexagonal networks. However, three-dimension networks with higher space-utilization rates are rarely exploited. On the other hand, the mechanical connection methods of TENG networks are only preliminarily studied. For complex ocean wave conditions, it is necessary to promote more comprehensive studies to construct TENG networks more effectively.

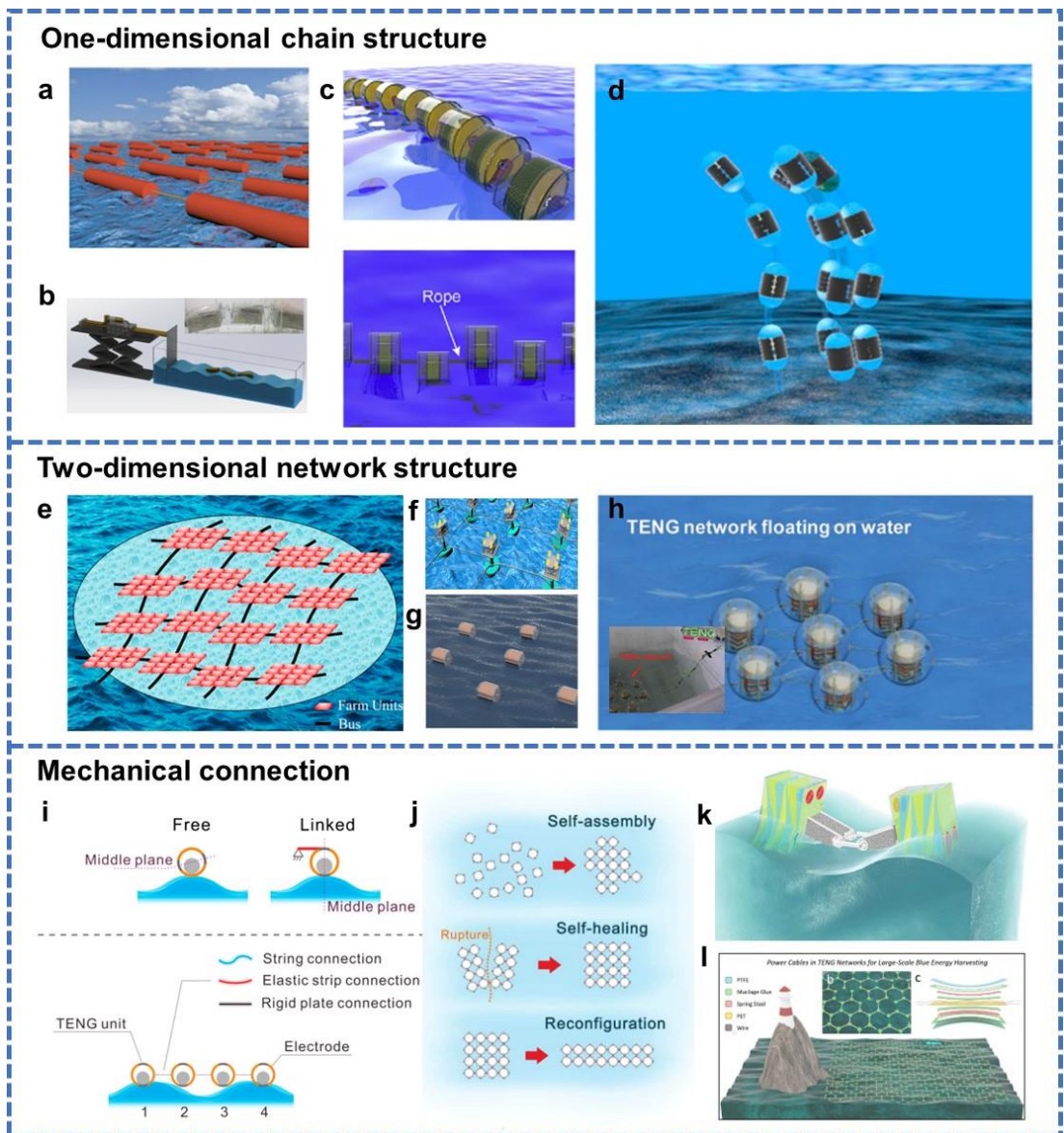

**Figure 3.** (**a**) Sea snake composed of TENG units. (**b**) Diagram of the TENG device floating on the water surface. (**a**,**b**) Reprinted with permission from ref. [51], Copyright 2022, Elsevier. (**c**) Structure of the tandem disk TENG device. Reprinted with permission from ref. [53], Copyright 2019, Elsevier. (**d**) Network connecting TENG units vertically to the bottom of the ocean. Reprinted with permission from ref. [39], Copyright 2019, American Chemical Society. (**e**) Configuration of the quadrilateral network with the multilayered connection. Reprinted with permission from ref. [54], Copyright 2015, American Chemical Society. (**f**) Square network of the buoy-based TENG device. Reprinted with permission from ref. [42], Copyright 2021, Elsevier. (**g**) Schematic diagram of the network composed of the soft-contact cylindrical TENG devices. Reprinted with permission from ref. [61], Copyright 2021, Wiley. (**h**) Hexagonal network containing seven TENG units. Reprinted with permission from ref. [56], Copyright 2019, Wiley. (**i**) Three types of connections: rigid plate connection, elastic strip connection, and string connection. Reprinted with permission from ref. [57], Copyright 2018, American Chemical Society. (**j**) Self-assembly TENG network with rational self-adaptive magnetic joints. Reprinted with permission from ref. [58], Copyright 2019, Elsevier. (**k**) Hybrid harvester designed by the pivot hinge. Reprinted with permission from ref. [59], Copyright 2019, Wiley. (**l**) TENG network based on the plane-like power cables. Reprinted with permission from ref. [60], Copyright 2020, the authors.

### 3.2. Circuit Connection of TENG Network for Water Wave Energy Harvesting

In the process of integrating TENG units into a network, the wiring of all units is fairly complicated; therefore, circuit connection is an important topic. Reasonable circuit-connection methods must have the ability to reduce the energy loss of the TENG unit's integration under irregular water waves. Furthermore, for the large-scale development and practical applications of the TENG network, the efficiency and cost of the circuit connection are also inevitable issues.

#### 3.2.1. Connection through Rectifier Bridges

In most of the current works, researchers applied rectifier bridges to connect TENG units in the networks because their motion states are difficult to be synchronous under the water waves. The rectifier bridges can not only convert the alternating current (AC) outputs of TENG units into the direct current (DC) outputs but also separate each unit electrically; therefore, the output energy can be superimposed without affecting each other. After connecting to rectifier bridges, since the output current of a single TENG unit is relatively small, the parallel connection is usually utilized to achieve the higher output current. For example, Xu et al. reported a square 4 × 4 network based on ball–shell TENG units in 2019 (Figure 4a), in which the sixteen units were rectified first, and then all of them were connected in parallel [57]. The authors tested the network with harmonic agitations and impact and found that the short circuit current shows a linear dependence on the number of units (Figure 4b). Similarly, Lin et al. also concluded the same regularity in their work on the TENG network of six units, namely that the output current increases linearly with the installed unit number increasing from one to six (Figure 4d), although the TENG units here have a completely different pendulum structure compared with the ball–shell structure (Figure 4c) [61]. In addition, the charging performance for different unit numbers was also measured in their work, and the charging rate also increased with the elevation of unit number, as demonstrated in Figure 4e. These works illustrate that the rectification is a very necessary step for TENG networking, and the output current and transferred charge of the TENG network can increase linearly with the unit number after rectification and parallel connection. At present, connecting TENG units in parallel through rectifier bridges is the simplest and most effective method.

#### 3.2.2. Connection through Charge-Excitation Scheme

Except the parallel connection through rectifier bridges, some researchers have developed other methods to complete the circuit connection of TENG networks. In 2019, based on a charge-excitation system with external charge excitation [62–64], Jiang et al. proposed a self-charge-supplement TENG network, which consists of one CS-TENG and six M-TENGs, as shown in Figure 4f [65]. The CS-TENG was designed as the pump TENG, which can supplement charges for the M-TENGs continuously to prevent the charge dissipation and the performance deterioration. Each unit of the M-TENGs contains three electrode layers and two dielectric layers, as shown in Figure 4g. The authors explored the influence of the connecting method between the CS-TENG and M-TENG. In Method 1, the top and middle electrodes of M-TENG are connected to the CS-TENG, while in Method 2, the lower two electrodes are connected. The two electrodes connected with the CS-TENG are charged by the CS-TENG through rectifier bridges, while the bottom and top electrodes serve as the output terminals. When the TENG network works in water waves, the potential difference between the electrodes drives the current flow from the top to the bottom layer through the external load, generating a periodic alternating current. The transferred charge and output voltage before and after the charge supplement by the two methods are shown in Figure 4h,i. The results illustrate that the introduction of the charge-supplement scheme significantly improves the output performance of the network, and that Method 2 is better. The authors ingeniously connected TENG units through the self-charge-supplement approach, which has the obvious merits of high stability and reliability, and being maintenance-free. In 2020, this research group designed a new charge-excitation circuit (CEC) specifically for the

hexagonal TENG network to harvest water wave energy (Figure 4k) [66]. The CEC can not only improve the output performance of the TENG units but also convert the AC output to the DC output at the same time. Therefore, for the TENG network proposed in this work, the seven TENG units were directly connected in parallel through the CECs without rectified bridges (Figure 4j). The output performance of the TENG network with the CECs under different water wave frequencies were systematically studied (Figure 4l,m). The maximum current of 24.5 mA was obtained at the wave frequency of 0.6 Hz, which is two orders of magnitude higher than the TENG network without CECs. The peak power of the charge-excitation TENG network reached 24.6 mW, and the power density was 6.71 W/m$^3$. Designing energy management or excitation modules with the function of rectification is a solution to improve the efficiency of the TENG unit integration.

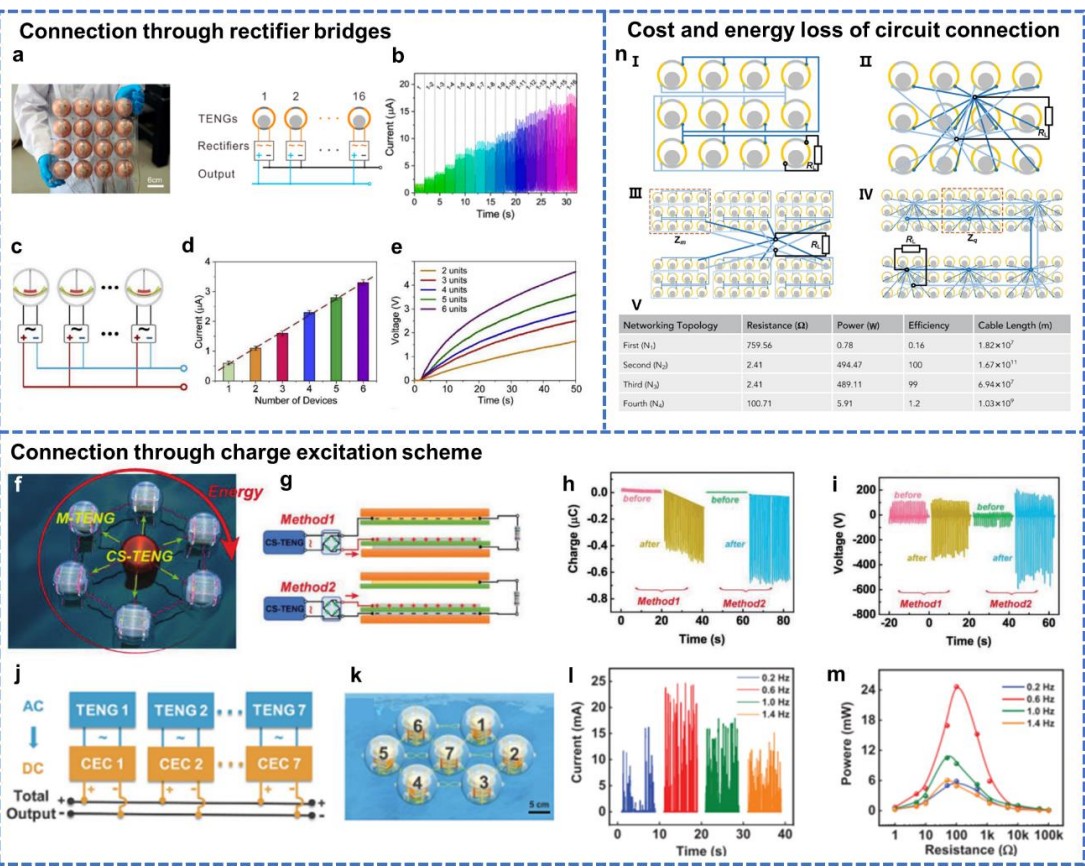

**Figure 4.** (**a**) Photograph and circuit connection of the square 4 × 4 network based on the ball–shell TENG unit. (**b**) Rectified short current of the network as the unit number increases from 1 to 16. (**a**,**b**) Reprinted with permission from ref. [57], Copyright 2018, American Chemical Society. (**c**) Schematic diagram of the rectification circuit for the network composed of the pendulum-inspired TENGs. (**d**) Output current and (**e**) charging performance for different unit numbers. (**c**–**e**) Reprinted with permission from ref. [61], Copyright 2019, Elsevier. (**f**) Self-charge-supplement TENG network. (**g**) Each unit of the M-TENGs containing three electrode layers and two dielectric layers. (**h**) Transferred charge and (**i**) output voltage for the TENG before and after the charge supplement by two methods. (**f**–**i**) Reprinted with permission from ref. [65], Copyright 2019, Willey. (**j**) Parallel connection of the units in the TENG network through the CECs without rectified bridges. (**k**) Diagram of the hexagonal TENG network integrated with the CECs. (**l**) Output current and (**m**) output power-resistance profiles of the charge-excitation TENG network for various water waves. (**j**–**m**) Reprinted with permission from ref. [66], Copyright 2020, Willey. (**n**) Four fundamental forms of electrical networking for large-scale TENG networks and list of merits and demerits. Reprinted with permission from ref. [67], Copyright 2020, Elsevier.

### 3.2.3. Cost and Energy Loss of Circuit Connection

Moreover, the cost and energy loss of the circuit connection are also important issues worthy of study in TENG networking, although little attention has been paid at present. For large-scale TENG networks, a huge number of cables are required, and their price cannot be ignored. On the other hand, since the output performance of the single TENG unit is relatively low, too-long cables with high resistance will severely limit the electrical energy output. In 2020, based on the typical spherical TENG, Zhang et al. systematically investigated four fundamental forms of electrical networking for large-scale TENG networks (Figure 4n) [67]. The first networking method connected the adjacent TENG units one by one (Figure 4n-I), while the second method directly connected the positive and negative poles of all TENG units to two points (Figure 4n-II). The other two methods involved multi-level circuit organization, which combined the first and second methods. For the third method, the TENG units were first connected by the first method and then connected by the second method (Figure 4n-III). The fourth method was the opposite of the third method (Figure 4n-IV). The authors systematically calculated the optimum load resistance, the maximum average power, the retention efficiency, and the cable length of the same TENG network (the number of units is $8.26 \times 10^7$) with the four connection methods (Figure 4n-V). With the second and third networking topologies, the TENG network can maintain high efficiency. However, the third topology is the better method because the efficiency only decreases by 1% while the cable length drops by three orders of magnitude. However, this work is only at the theoretical level, and experiments to analyze large-scale TENG networks need to be carried on to prove the theory.

At present, there is still a lack of systematic research on the circuit-connection method of the TENG network. For most of the TENG networks, the rectifier bridges are applied to connect TENG units in parallel. Additionally, the charge-supplementary systems and charge-excitation schemes are designed in some works to realize the circuit connection of the TENG networks. In order to advance the practical applications of the TENG network, the effective wiring methods need further experiments.

### 3.3. Output Performance of TENG Network

In Table 1, the output performances of some TENG networks reported in recent years are summarized. The two-dimensional planar structure was adopted the most, and the number of units was several or more than a dozen. The rectifier bridges were commonly utilized in the circuit connection, and the overall peak power of TENG networks were all on the order of milliwatts. Moreover, it can be seen that the power density is closely related to the volume and number of TENG units. As the volume and number of TENG units increases, it becomes more difficult to improve the power density of the whole TENG network. Therefore, how to expand the scale of the TENG network while maintaining high power density is a problem. For the charging speed, designing a charge-excitation scheme was proven to be a significant way to improve the charging speed of the TENG networks. Focusing on different aspects, the researchers in these works carried out a preliminary exploration of TENG networking.

**Table 1.** Summary of typical TENG networks.

| Network Topology | Unit Structure | Unit Number | Unit Size | Circuit Connection | Peak Power (mW) | Peak Power Density (W/m³) | Charging Speed | Test Environment | Year | Refs. |
|---|---|---|---|---|---|---|---|---|---|---|
| Square | Spring-assisted multilayered structure | 4 | Spherical unit diameter 10 cm | Rectifier bridges | 15.97 | 7.62 | 120 s 0.74 V (1 mF) | Water waves | 2018 | [37] |
| Square | Ball-shell structure | 16 | Spherical unit diameter 7 cm | Rectifier bridges | 12.84 | 4.47 | 23 s 3.00 V (44 μF) | Impact agitation | 2018 | [57] |
| Chain | Sea snake structure | 3 | Cuboid unit length 6.4 cm, width 5.1 cm, height 10.1 cm | Rectifier bridges | - | - | 600 s 2.00 V (100 μF) | Water waves | 2018 | [51] |

**Table 1.** *Cont.*

| Network Topology | Unit Structure | Unit Number | Unit Size | Circuit Connection | Peak Power(mW) | Peak Power Density(W/m³) | Charging Speed | Test Environment | Year | Refs. |
|---|---|---|---|---|---|---|---|---|---|---|
| Rectangle | Pendulum structure | 6 | Spherical unit diameter 12 cm | Rectifier bridges | - | - | 780 s 3.30 V (100 μF) | Water waves | 2019 | [61] |
| Square | Torus structure | 16 | Torus unit major diameter 7 cm | Rectifier bridges | - | - | 300 s 1.50 V (47 μF) | Water waves | 2019 | [68] |
| Rectangle | 3D electrode structure | 18 | Spherical unit diameter 8 cm | Rectifier bridges | 34.60 | 7.16 | 21 s 1.33 V (216 μF) | Water waves | 2019 | [58] |
| Hexagon | Spring-assisted multilayered structure | 7 | Spherical unit diameter 10 cm | Rectifier bridges | 12.20 | 3.33 | 60 s 2.54 V (1 mF) | Water waves | 2019 | [56] |
| Rectangle | Swing structure | 6 | Cylindrical unit diameter 10 cm, thickness 5 cm | Rectifier bridges | - | - | 200 s 2.20 V (100 μF) | Water waves | 2020 | [31] |
| Hexagon | Spring-assisted multilayered structure | 7 | Spherical unit diameter 10 cm | Charge excitation scheme | 24.60 | 6.71 | 30 s 1.80 V (4.7 mF) | Water waves | 2020 | [66] |
| Square | Spring-assisted swing structure | 4 | Spherical unit diameter 12 cm | Charge excitation scheme | 16.64 | 4.60 | 30 s 1.80 V (2.2 mF) | Water waves | 2021 | [49] |

## 4. Summary and Perspectives

In this review, various aspects of the TENG units' network evolution are systematically summarized. Firstly, integrating TENG units in one point-absorption device is the fundamental step of constructing the TENG network. The most widely utilized topology organization methods are stacking integration and multidirectional integration. Secondly, the circuit connection of the TENG units in one point-absorption device is also an important issue. Using a direct connection and utilizing rectifier bridges are the most common methods; however, designing the charge-supplement system is a better way. Thirdly, the topology structures of the TENG network are summarized. The one-dimension chain structures are generally horizontal and vertical, and the two-dimension plane structures are quadrilateral and hexagonal. Finally, the circuit-connection schemes of the whole TENG network are analyzed. The rectifier bridges are often applied to connect each unit in parallel, and the charge-supplementary systems and charge-excitation schemes are developed in some works. For the realization of carbon neutrality, developing clean and renewable ocean wave energy by TENG networks has great application prospects. However, the current state of TENG network research is inadequate for real industrialization. The following are the existing problems in the field of TENG network development, as well as suggested future research directions:

1. Although some experimental studies on the topology structure of the TENG network have been carried out, the specific theoretical guidance is still lacking. Therefore, the mechanical model of the interaction between ocean water waves and the TENG networks is worth investigating. The theoretical model can effectively optimize the topological organization of the TENG networks;

2. Current experiments of the TENG network are conducted under laboratory conditions, and it is difficult to simulate the real marine environment, such as water salinity and corrosion, and random wave frequency and amplitude. Solving the potential problems when the TENG networks work under the ocean waves is crucial to its commercialization;

3. The complete linear superposition of the output performance of each unit in the TENG networks is difficult to achieve due to the energy consumption of the components, such as rectifier bridges and wires. Combining the recently proposed charge-supplement or energy-excitation schemes with the TENG networks for water wave energy harvesting is a feasible method to improve the energy-conversion efficiency;

4.  Regarding the commercialization of the TENG network, how to integrate the electrical energy generated by the TENG network into the commercial power gird is a considerable question. The output power of the TENG network under irregular ocean waves must be converted into high-voltage AC power, which is usually required for long-distance electrical energy transmission. Since the development of TENG networks is still at an exploratory stage, this problem has been rarely discussed;

5.  Cost control is also an important issue for the engineering applications of the large-scale TENG network, although researchers usually do not consider cost issues in the experimental studies. Cost control may be conducted from various perspectives, such as material selection, circuit simplification, and cable reduction.

**Author Contributions:** Conceptualization, X.L. and T.J.; original draft preparation, X.L.; review and editing, S.L., H.Y. and T.J.; supervision, T.J.; funding acquisition, T.J. and X.L. All authors have read and agreed to the published version of the manuscript.

**Funding:** Project supported by Fundamental Research Funds for the Central Universities (E2E46805); China National Postdoctoral Program for Innovative Talents (BX20220292); China Postdoctoral Science Foundation (2022M723100); and the Innovation Project of Ocean Science and Technology (22-3-3-hygg-18-hy).

**Data Availability Statement:** Not applicable.

**Conflicts of Interest:** The authors declare no conflict of interest.

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
