# Peer review of "Triboelectric Nanogenerators for Ocean Wave Energy Harvesting: Unit Integration and Network Construction"

_electronics, doi:10.3390/electronics12010225_

Round 1

Reviewer 1 Report

This work systematically summarizes the recent research advances on TENG development for ocean wave energy harvesting. The focuses of this work are TENG unit integration and networking. The topic falls in the scope of Electronics. The content is well organized, and meaningful suggestions and perspectives are brought up by the authors. I would like to recommend this work to be accepted by Electronics after some minor revisions. 

1. English needs to be further polished.

2. A new title is suggested for the authors' consideration, "Triboelectric Nanogenerators for Ocean Wave Energy Harvesting: Unit Integration and Network Construction".

3. Power density (W/m2 or W/m3) is an effective figure of merit of TENGs harvesting ocean wave energy. Please consistently use power density if possible for the comparison across different work. Power densities were used in some places while power in others. 

4. Please provide device general dimensions in the text if possible. It will help readers to get a better sense of the TENG devices. Are they tested with the simulated ocean wave in lab or actual ocean wave? 

Reviewer 2 Report

In this work, the ocean wave energy harvesting through the triboelectric nanogenerators has been summarized in all aspects. Especially the topology of design and circuit connection scheme are systematically reviewed.

General comment: This work provides a weak data arrangement to convince what the authors report. The authors need to provide more proper comparison data related to TENG structure or circuit connection effect to TENG units/network output performance. Please follow the comments below for revision. I recommend accepting this review article after satisfying the below comments.

Major reviews:

-          Related to the content of Figure 1 for structure of TENG units integrated in a point absorption device for blue energy harvesting, the authors need to describe the working mechanisms for each structure briefly, not only introduce.

-          Why figure 1 is divided into 3 different groups, please clearly the group name through Figure 1 caption.

-          Figure 2 is about the circuit connection of TENG units integrated in a point absorption device. However, the figure arrangement is not well prepared. Please clearly show the circuit connection of TENG units following the order as: schematic (large enough, not small as the size of the TENG unit in Figure 2c) – circuit connectionoutput performance with different number of TENG units. All should be unified.

-          Clarify the 3 groups figure in Figure 2, it can be added the kind of circuit connection in figure caption.

-          In Figure 3, the authors need to show the advantages and disadvantages for each arrangement method, specifically with 1D chain structures or 2D plane structures more deeply.

-          In Figure 4, the circuit connection and output data should be compatible. For example, Figure 4b shows about 16 units, but output data in Figure 4(d-e) are with 6 units only. In addition, the rest circuit connection effects output performance of other designs is not described. Please add it.

-          At least one more figure is needed to describe the effect of each topology structure or circuit connection of TENG units/networks and the output performance data. Through these, the challenges, circumstance, and perspectives can be well organized.

Minor reviews:

-          In some symbol notations, it is needed to be set the superscript for the number, for example, some symbols in line #58, 113-114, etc.

-          The Figure 1 is about the topology of TENG units, please remove the output power density in Figure 1c and output current/voltage in Figure 1i

Round 2

Reviewer 2 Report

The authors have provided their well responses. In overall, the manuscript can be accepted to be published after major #6 is revised well.

In Figure 4, the circuit connection and output data should be compatible, the circuit connection effects output performance of other designs is not described for "Cost and energy loss of circuit connection" (Fig. 4j), and "Connection through charge excitation scheme" (Fig. 4f-i). Please add it.

Round 3

Reviewer 2 Report

It is well revision. I recommend this manuscript to be published.

Thank you.